# An Analysis of Primary Hyperparathyroidism in Association with Depression or Anxiety

**DOI:** 10.3390/diseases13020054

**Published:** 2025-02-12

**Authors:** Ana-Maria Gheorghe, Claudiu Nistor, Aurelian-Emil Ranetti, Mara Carsote

**Affiliations:** 1PhD Doctoral School of “Carol Davila” University of Medicine and Pharmacy, 020021 Bucharest, Romania; ana-maria.gheorghe@drd.umfcd.ro; 2Department 4-Cardio-Thoracic Pathology, Thoracic Surgery II Discipline, “Carol Davila” University of Medicine and Pharmacy, 050474 Bucharest, Romania; 3Thoracic Surgery Department, “Dr. Carol Davila” Central Military University Emergency Hospital, 010242 Bucharest, Romania; 4Department of Endocrinology, “Dr. Carol Davila” University of Medicine and Pharmacy, 020021 Bucharest, Romania; carsote_m@hotmail.com; 5Endocrinology Department, “Dr. Carol Davila” Central Emergency University Military Hospital, 010825 Bucharest, Romania; 6Department of Clinical Endocrinology V, “C.I. Parhon” National Institute of Endocrinology, 011863 Bucharest, Romania

**Keywords:** primary hyperparathyroidism, depression, anxiety, PTH, calcium, parathyroidectomy, parathyroid, hormone, scale, questionnaire, study

## Abstract

Background: Non-classical manifestations such as neuropsychiatric manifestations in primary hyperparathyroidism (PHPT) have long been documented as symptoms of PHPT and are commonly reported by these patients, despite this connection still being a matter of debate, and they (*per se*) do not represent an indication of parathyroidectomy. Objective: We aimed to overview the most recent findings regarding the link between depression and/or anxiety (D/A) in subjects confirmed with PHPT, including the impact of the surgery in improving their outcome. Methods: This was a comprehensive review of English-based original studies published between January 2020 and October 2024. Results: The studies (n = 16) included a total of 10,325 patients and an additional 152,525 patients with hypercalcemia (out of whom 13,136 had a PHPT diagnosis and 45,081 were at risk of PHPT diagnosis). Out of these subjects with PHPT, 10,068 underwent parathyroidectomy. Female prevalence was between 62.5 and 92%. Most individuals were over 50, with the youngest studied population having a mean age of 52.7 ± 13.8 years, and the oldest had a median of 71. Depression was documented based on ICD-10 codes (n = 3) and patients’ records (n = 2), Depression Anxiety Stress Scales (DASS) (n = 2), Beck Depression Inventory (BDI) (n = 3), BDI-II (n = 3), Symptom Check List 90-revised (SCL) (n = 1), Hamilton Depression Rating Scale (HAM-D) (n = 2), HADS (n = 2), Patient Health Questionnaire-9 (n = 1), and European Quality of Life 5 Dimensions 3-Level Version (EuroQOL-5D-3L) (n = 1). Patient records’ (n = 1) and ICD-10 codes (n = 2) were also used for anxiety. Most studies used questionnaires to identify anxiety in PHPT: DASS (n = 2), SCL90R (n = 1), Generalized Anxiety Disorder-7 (n = 1), HADS (n = 2), EuroQOL-5D-3L (n = 1), and State–Trait Anxiety Inventory (n = 1). Depression prevalence varied from 20–36.6% to 65.7% (scale-based assessment) and to 10.5% upon ICD-10. A rate of newly onset depression was reported of 10.7% and of 0.2% with concern to the prevalent suicidal ideation (an incidental rate of 0.4% after a median follow-up of 4.2 years). Most studies identified a moderate depression (when assessing its severity), affecting approximately one third of the surgery candidates. The prevalence of anxiety in PHPT varied between 10.4% and 38.6% (n = 8). Discordant results were generated when applying distinct questionnaires for the same population, and this might come as a potential bias. Other confounding factors are generated by the sub-population referred for surgery that typically displays a more severe parathyroid condition or non-endocrine overlapping conditions (e.g., related to the social or familial status). Conclusion: The modern approach of the patient with PHPT should be complex and go beyond the traditional frame. D/A had a high prevalence in the mentioned studies, associated with increased medication use. Yet, the underlying pathogenic mechanisms remain incompletely elucidated. No correlations between D/A and serum calcium levels were confirmed, while PTH had a slight positive correlation with depression. Parathyroid surgery appears to be beneficial for D/A as it improves the scores, prevalence, and severity. Cinacalcet might reduce depression scores, although more evidence is needed. Women are prone to both PHPT and D/A. The optimal method of D/A screening in PHPT remains to be determined, and the current scales need validation and perhaps adjustment for this specific population sub-group, while PHPT management should be refined upon D/A identification.

## 1. Introduction

Primary hyperparathyroidism (PHPT) is a complex disease with various clinical consequences beyond the classical bone and the kidney involvement [1,2]. Non-classical manifestations such as neuropsychiatric manifestations have long been documented as symptoms of PHPT and are commonly reported by these patients, despite this connection still being a matter of debate [3,4]. The underlying pathogenic pathways linking hypercalcemia and parathyroid hormone (PTH) excess to the co-presence of the neuropsychiatric symptoms are poorly understood [5]. The most common psychiatric manifestations in patients with PHPT are depression and anxiety, with a prevalence up to 30% and 49%, respectively, as reported by older studies [6,7]. Of note, fatigue, sleep disturbances, and even psychosis have been reported in PHPT, too [8,9].

Depression is accompanied by a wide spectrum of vegetative, emotional, and cognitive symptoms, including fatigue, appetite, weight fluctuations, insomnia, and anhedonia, all of which impair physical and emotional health, as well as social well-being [10]. Similarly, the psychological and physical symptoms occurring in patients with anxiety, including insomnia, fatigue, muscle tension, and attention difficulties, can be debilitating. Anxiety impairs one’s overall health, it increases the risk of cardiovascular events, and affects work productivity as well as the personal life [10,11]. Both anxiety and depression decrease the quality of life and increase the disease-related overall burden [12]. Moreover, they increase the associated healthcare costs due to medication and additional therapy, and they may impact the evolution of other comorbidities in PHPT such as osteoporosis/rehabilitation upon osteoporotic fractures, kidney ailments, and cardiovascular diseases [13,14,15,16,17]. Moreover, since the association between PHPT and depression/anxiety is not the norm, the acumen of differential diagnosis involves an overlapping diagnosis with the parathyroid tumour that might not be directly related to the calcium and/or PTH over-production, or it may actually be related to other non-endocrine ailments or even concurrent medications [10,18,19].

Identifying depression and anxiety in patients with PHPT may be challenging. Their screening is recommended, as seen in the general adult population, considering that early detection and associated treatment improve the outcome. There are various scales available and validated, such as Hospital Anxiety and Depression Scales (HADS), Patient Health Questionnaire-9 (PHQ-9), Generalized Anxiety Disorder (GAD), etc. [20,21]. While one of the most reliable tools seems to be the PHQ-9 [21,22,23], there is no standardized approach recommended and no distinct recommended instrument so far (at guideline level) [20]. Regarding PHPT patients, recent data suggested that in PHPT, the PHQ-9 score may be applied in surgery candidates since parathyroidectomy was shown to reduce the preoperatory scores and decrease the prevalence of clinically significant depression [24].

According to the American Association of Endocrine Surgeons, patients with psychological manifestations such as depression and anxiety might have indication for parathyroidectomy, as it could improve the outcome. However, the evidence in support of this claim is still scarce [25]. So far, there are no specific recommendations regarding the diagnosis and management of depression in patients with PHPT or to perform surgery if one patient presents this type of complication (unless the other traditional complications are co-presented, such as osteoporosis, kidney stones and/or failure, high levels of calcium, etc.) [26].

We aimed to overview the most recent findings regarding the link between depression and/or anxiety in subjects confirmed with PHPT, including the impact of the parathyroidectomy in improving the outcome of these conditions; finally, we provided a novel interdisciplinary framework.

## 2. Methods

This was a comprehensive review of the English-based medical literature. We searched for papers published between January 2020 and October 2024, using the following keywords: “depression”, “anxiety” in different combinations with “hyperparathyroidism”, “normocalcemic hyperparathyroidism”, “hypercalcemic hyperparathyroidism” or “asymptomatic hyperparathyroidism”. The search based on the mentioned keywords identified 929 articles, the time frame restricted the search, and, finally, the articles were manually checked within title and abstract to provide original studies in the mentioned topics. We only included clinically relevant, original studies regarding depression and/or anxiety (including associated medication) in individuals confirmed with PHPT. We excluded papers on children, adolescents, pregnant women, editorials, reviews, meta-analyses, secondary (renal or tertiary) hyperparathyroidism, duplicates, non-English articles. Overall, we included 16 studies [27,28,29,30,31,32,33,34,35,36,37,38,39,40,41,42] (Figure 1).

## 3. Results

The studies (n = 16) [27,28,29,30,31,32,33,34,35,36,37,38,39,40,41,42] included a total of 10,325 patients diagnosed with PHPT and an additional 152,525 patients with hypercalcemia (out of whom 13,136 had a PHPT diagnosis and 45,081 were at risk of PHPT diagnosis). Out of the subjects with PHPT, 10,068 underwent parathyroidectomy, and the analysis was performed with respect to the pre- and postoperatory panel in surgery candidates [27,28,29,30,31,32,33,34,35,36,37,38,39,40,41,42] (Table 1).

The majority of the patients were women (between 62.5% [27] and 92% [28]). Most individuals were over 50 years, with the youngest studied population having a mean age of 52.7 ± 13.8 years [42] and the oldest with a median age of 71 (62–79) years [37]. Depression was documented based on ICD-10 (The International Classification of Diseases-Tenth Revision Clinical Modification) codes (n = 3) [27,30,34] and patients’ records (n = 2) [33,42], as well as upon applying the following questionnaires: Depression Anxiety Stress Scales (DASS) (n = 2) [28,32], Beck Depression Inventory (BDI) (n = 3) [32,36,40], BDI-II (n = 3) [29,31,41], Symptom Check List 90-revised (SCL90R) (n = 1) [32], Hamilton Depression Rating Scale (HAM-D) (n = 2) [35,41], HADS (n = 2) [37,41], Patient Health Questionnaire-9 (n = 1) [38], and European Quality of Life 5 Dimensions 3-Level Version (EuroQOL-5D-3L) (n = 1) [39]. Patient records (n = 1) [33] and ICD-10 codes (n = 2) [30,34] were also used for anxiety. Most studies, however, used questionnaires to identify anxiety in patients with PHPT: DASS (n = 2) [28,32], SCL90R (n = 1) [32], Generalized Anxiety Disorder-7 (n = 1) [35,38], HADS (n = 2) [37,41], EuroQOL-5D-3L (n = 1) [39], and State–Trait Anxiety Inventory (n = 1) [40].

### 3.1. Analysis of Depression in Patients Diagnosed with Primary Hyperparathyroidism

#### 3.1.1. Depression: Prevalence, Severity, and Influencing Factors

The prevalence of depression in patients with PHPT varied with the studies, noting that distinct tools were used to assess it. The highest prevalence was 65.7% in the study conducted by Jovanovic et al. [32], BDI being used in this instance [32]. BDI-II was also applied in two other studies that identified depression rates of 57.1% and 51%, respectively [29,31]. The lowest prevalence (of 36.6%) based on BDI-II was reported by Kunert et al. [41]. The same study found a higher prevalence (of 60.4%) when HAM-D was applied and a lower prevalence of 20.79% upon HADS use, respectively [41]. Overall, the lowest rate was found at 10.5% based on ICD-10 codes. The same study also reported newly onset depression in 10.7% of the patients with parathyroid tumours, prevalent suicidal ideation in 0.2% of them, and newly onset suicidal ideation in 0.4% of these subjects after a median follow-up of 4.2 (3.7–4.6) years [30]. Another study based on the patients’ records identified that 27.2% of them had depression, women being more often affected (*p* = 0.016) [42].

Some studies provided depression scores in patients with PHPT. For instance, Wang et al. [40] found a median score of 5 (0.46) using BDI-II [40], while Chan et al. [28] reported a mean (SE) score of 6.89 (0.95) using DASS [28], both corresponding to the normal population. Similarly, Komman et al. [34] reported median scores of 4 (1–6) and 4 (1–8) in patients with PHPT ≥ 50 years and ≥70 years, respectively [34].

The variable results in the same study population due to using different methods/tools of depression assessment are worth mentioning. Two of the studies applied multiple methods within the same study population. For example, Kunert et al. [41] identified discordant rates via HAM-D, BDI-II, and HADS in 101 patients with PHPT who underwent parathyroidectomy and 50 controls with toxic goitre. At baseline, in patients with PHPT, the overall prevalence of depression was the highest according to HAM-D (60.4%) and the lowest according to HADS (20.79%). Moreover, when HAM-D and BDI-II were used, the prevalence of depression in PHPT patients was higher than controls (60.4% versus 12%, *p* < 0.001 and 36.36% versus 18%, *p* < 0.05, respectively), but the difference was not statistically significant when HADS was applied (20.79% versus 12%, *p* > 0.05) [41]. BDI and DASS were applied by Jovanovic et al. [32] in 101 individuals with PHPT who underwent surgery and found a higher rate of depression based on BDI (65.7%) versus DASS (19.8%) [32]. Compared to the control population, depression was more frequent and severe in patients with PHPT [31,38,41], while versus age- and sex-matched controls, the endocrine subjects with PHPT had a higher BDI-II score (16.8 ± 9.98 versus 6.10 ± 5.45, *p* < 0.001) [29]. Similar results were found between patients with PHPT and those diagnosed with a toxic goitre [41]. Another study compared subjects who underwent parathyroidectomy to those who were referred for thyroidectomy and revealed that subjects with parathyroid tumours had higher preoperative depression scores (6.7 ± 6.6 versus 4.4 ± 4.9, *p* < 0.01) [38].

Of note, the prevalence of depression might be influenced by the inclusion of surgical patients only, who typically display a more severe disease form [29,32,41]. Yet, Song et al. [30] analysed operative and non-operative subjects and found similar rates with respect to depression (*p* = 0.265), pre-existing suicidal ideation (*p* = 1), and newly onset suicidal ideation (*p* = 0.13) [30]. The study reported higher rates of newly onset depression in non-operative patients (12.2%) compared to the overall study population (10.7%) and operative patients (9.3%) (*p* = 0.046) [30].

Other factors influencing the depression rates were investigated by Febrero et al. [31], who identified higher depression scores in subjects aged between 40 and 60 years (*p* < 0.001), those with no primary education (*p* = 0.014), and individuals with children (*p* = 0.019). Regarding PHPT-related symptoms, depression rates were similar in patients experiencing bone pain (*p* = 0.54), nephrolithiasis (*p* = 0.215), gastrointestinal complains (*p* = 0.863), and hypertension (*p* = 0.323) [31].

Major depressive disorder was also evaluated in a very large study on 135,034 patients diagnosed with hypercalcemia (and 13,136 individuals had PHPT), the condition being more prevalent in subjects with PHPT as well as those at high risk of PHPT diagnosis with or without PTH data when compared to matched controls (*p* < 0.001). Moreover, the time to diagnosis might influence the development of major depression. The prevalence and risk of major depression were lower in subjects diagnosed with PHPT later (after more than one year from the moment of hypercalcemia confirmation) compared to those diagnosed earlier than one year [11.9% versus 16.1%, odds ratio (OR) = 0.7 (0.62–0.8), *p* < 0.001], but, overall, during the following one to three years, depression became more prevalent in patients diagnosed later with PHPT [17.3% versus 15.1%, OR = 1.18 (1.05–1.34), *p* = 0.008)] [33]. On the other hand, the identification of depression may influence the odds of PTH testing in patients with hypercalcemia, as suggested by Bunch et al. [27] in a large retrospective study (N = 17,491). The people suffering from depression and hypercalcemia had lower odds of having a PTH check-up [OR = 0.84 (0.74, 0.96), *p* = 0.0081], suggesting a potential testing gap on this specific matter [27] (Table A1).

#### 3.1.2. The Use of Antidepressant Medication in Subjects with Primary Hyperparathyroidism

The prevalence of antidepressant use in PHPT was reported at 20% in surgery candidates [29]. The largest study investigating the matter of this drug administration was performed on 8279 subjects with PHPT [37]. The most common drugs were selective serotonin reuptake inhibitors (SSRIs) in 15.9% of the subjects, followed by selective serotonin and norepinephrine inhibitors (SNRIs) in 8.6%, while 4.5% of the people with PHPT were treated with tricyclic antidepressants. Compared to controls, the odds of using antidepressants were higher in subjects with PHPT, for SSRI [OR = 1.38 (1.3–1.47)], SNRI [OR = 1.36 (1.26–1.48)], and tricyclic antidepressants [OR = 1.4 (1.26–1.57)] [34] (Table A2).

#### 3.1.3. The Relationship Between Depression and the Endocrine Assays Amid Primary Hyperparathyroidism

Six studies investigated potential associations between the presence of depression and the levels of calcium, parathyroid hormone, or other endocrine parameters such as serum cortisol and bone formation marker osteocalcin in PHPT [30,31,34,38,40,41]. Regarding the baseline serum calcium levels, no statistically significant relationship with depression was found. Song et al. [30] calculated a threshold of 11.5 mg/dL for the value of serum total calcium but found no statistical significance in terms of hazard ratio (HR) for depression [HR = 1.32 (0.97–1.81)]. However, patients with serum total calcium above 11.5 mg/dL were 6.69-times more likely to develop suicidal ideation [HR = 6.69 (1.2–37.35)] [30]. A lower serum total calcium threshold was applied by Febrero et al. [31] while comparing the depression scores in patients with preoperative serum calcium lower than 11 mg/dL versus those with a calcaemic level ≥11 mg/dL. Both groups had similar depression scores (15.5 ± 11.38 versus 14.5 ± 19, *p* = 0.399) [31]. Koman et al. [34] used different serum calcium ranges in order to assess the medication use in PHPT compared to controls. The study found the highest odds for SSRI use [OR = 1.61 (1.39–1.99)] and SNRI use [OR = 1.81 (1.53–2.14)] compared to controls at lower calcium levels (<1.38 mmol/L), while calcaemic values did not influence the odds of tricyclic antidepressive use (*p* = 0.1771) [34]. Notably, the correlations between depression and pre- and postoperative calcium levels [38], depression and calcaemic levels [40], or depression severity and calcium levels [41] did not confirm any association (Table A3).

#### 3.1.4. The Impact of the Parathyroidectomy on Depression

The impact of parathyroid tumour removal on depression was studied in five original studies that analysed the presence of the pre- and postoperative depression scores in 448 surgery candidates [28,29,32,36,38]. Moreover, one study provided comparative data between parathyroidectomy and cholecystectomy [35], and another investigated the impact of parathyroidectomy in symptomatic and asymptomatic patients [39], while another analysed the antidepressant use before and after parathyroid surgery [34]. Most studies confirmed the depression prevalence decrease, as well as improvements in the depression scores and severity following parathyroidectomy. The largest study included 244 subjects with PHPT and 161 controls undergoing thyroidectomy. A higher rate of patients without depression following surgery (72.5% versus 47.9%) was confirmed, as well as an overall reduction in the mean depression scores after surgery (6.7 ± 6.6 versus 3.1 ± 3.9, *p* < 0.01). In addition, there was a decrease in depression severity based on a lower postoperatory proportion of moderate to severe depression (27.5% versus 8.2%, *p* < 0.01). Moreover, the study reported higher preoperative depression scores in PHPT compared with the subjects who underwent thyroidectomy (6.7 ± 6.6 versus 4.4 ± 4.9, *p* < 0.01), with similar postoperative scores following parathyroidectomy and thyroidectomy (3.1 ± 3.9 versus 3.3 ± 4.1, *p* = 0.52) [38].

An improvement in both depression scores and depression severity was also reported in a prospective study on 49 patients with PHPT who underwent parathyroidectomy compared with age- and sex-matched controls. At diagnosis, 42.9% of them did not have depression, but, one year after surgery, 72.9% had normal scores, suggesting the absence of depression. Not only was the prevalence of depression lower after parathyroidectomy but also the mean depression scores decreased, both at three months and one year after surgery compared to baseline (13.08 ± 10.76 versus 16.80 ± 9.98, *p* = 0.001, and 10.5 ± 10.79 versus 16.65 ± 10.03, *p* < 0.001, respectively). Despite score decreases at three months and one year after surgery, they remained higher than controls (13.08 ± 10.76 versus 6.10 ± 5.45, *p* < 0.001 and 10.5 ± 10.79 versus 6.10 ± 5.45, *p* < 0.001, respectively). Data provided by this study suggested that parathyroidectomy has a beneficial effect on depression in individuals confirmed with PHPT, while patients with PHPT may be more affected by depression than the healthy population [29].

The depression diagnosis was heterogeneous considering various scales. One study reduced this bias by using three different questionnaires, all showing an improvement in scores, suggesting an overall beneficial post-surgery effect. The study was conducted in 101 patients who underwent parathyroidectomy and showed a reduction in depression scores following surgery of up to 62.7%. The highest score reduction was observed when SCL90R was used, while the lowest reduction (of 56.2%) was confirmed based on DASS. According to BDI, there was a 60% reduction in the mean depression score following surgery. All scales showed lower scores after one month and six months after parathyroid surgery compared to preoperative scores, as well as lower scores six months post-surgery compared to one month after parathyroidectomy. Of note, the differences between questionnaires were supported by the fact that distinct scores provided distinct depression prevalence in the same study population. While BDI reported 65.7% of patients as having depression, DASS showed that only 19.8% had the condition. In addition, the mean preoperative BDI score (13.5) corresponded to depression, while the mean DASS depression score (6.3) corresponded to normal ranges [32].

While most studies found an improvement regarding depression symptoms following parathyroidectomy, Szalat et al. [36] did not confirm it at one month or six months following surgery (*p* = 0.697); of note, this was a small cohort (N = 18) [36]. When compared with other surgical procedures such as cholecystectomy, parathyroidectomy might be associated with more depressive symptoms in surgery candidates, as suggested by a retrospective study in 43 patients (parathyroidectomy candidates) and another 43 (cholecystectomy candidates). Subjects who underwent parathyroid surgery had higher median depression scores (via HAM-D) compared with those who underwent cholecystectomy at baseline (6 versus 2, *p* < 0.05), six months postoperatively (2 versus 1, *p* = 0.006) and one year post-surgery (2 versus 1, *p* = 0.007), respectively. However, all median scores corresponded to the normal ranges [35].

Medication against depression was analysed in a large study in 8279 parathyroidectomy candidates. The antidepressant use before and after surgery showed that SSRI use increased overall [relative risk (RR) at 0–6 months postoperatively = 1.25 (1.16–1.34) and RR at 7–12 months postoperatively = 1.28 (1.2–1.37)] but decreased in prevalent users [RR at 0–6 months postoperatively = 0.92 (0.87–0.97), RR at 7–12 months postoperatively = 0.94 (0.89–0.99)], with no statistically significant trend regarding the time frame [34]. The impact of parathyroidectomy in symptomatic and asymptomatic subjects was evaluated in a prospective study (N = 78 patients with PHPT who underwent parathyroidectomy, 28 of whom were asymptomatic, with the remaining having symptomatic PHPT). The study assessed depression and anxiety together. Overall, the 41% decrease in anxiety/depression was not statistically significant (*p* = 0.07), while asymptomatic patients had a similar anxiety/depression prevalence before and after surgery (*p* = 1.00), and symptomatic PHPT subjects had a reduction in the prevalence of moderate anxiety/depression postoperatively (25% versus 53%, *p* < 0.01) but no difference in extremely severe anxiety/depression prevalence (4% versus 4%) [39].

To summarize, these recent data suggested that parathyroidectomy improved the subjective depression symptoms and reduced depression prevalence (Table A5).

#### 3.1.5. The Administration of Cinacalcet and Its Impact on Depression

The use of cinacalcet was reported in one prospective observational study on 35 adults with PHPT and mild cognitive impairment who underwent treatment with this drug before surgery. At 4 weeks of cinacalcet versus baseline, depression scores decreased [−1 (0, −2), *p* = 0.019], as well as depression and anxiety scores combined [−2 (0, −3), *p* = 0.020] via HDAS. The reduction was not statistically significant in the sub-group of adults aged 70 years and older (*p* = 0.053 and *p* = 0.095, respectively). Parathyroidectomy improved the depression scores further in the overall population [−3 (0, −6), *p* = 0.04]. Of note, the baseline median depression scores corresponded to no or few symptoms [37] (Table A5).

#### 3.1.6. Depression in Relationship with the Conservative Versus Surgical Management of Parathyroid Tumours

Data regarding the impact of parathyroidectomy compared to the conservative approach were provided in a retrospective study, which included, after a propensity score match, 1918 patients with PHPT, out of who 959 underwent surgery and 959 were managed conservatively. The study reported a higher prevalence of newly onset depression in conservatively managed subjects versus those who underwent surgery (12.2% versus 10.7%, *p* = 0.046). However, there was no statistically significant difference in terms of pre-existing depression, pre-existing or newly onset suicidal ideation between the two groups, respectively. Moreover, the risk of depression and suicidal ideation was also similar in conservatively versus surgically managed subjects [30] (Table A6).

### 3.2. Analysis of Anxiety in Subjects Confirmed with Primary Hyperparathyroidism

#### 3.2.1. The Prevalence and Severity of Anxiety

The prevalence of anxiety in patients with PHPT varied between 10.4% [30] and 38.6% [41], and this variability might be caused by different methods to assess anxiety across eight studies [28,30,32,33,34,38,40,41]. For instance, the lowest prevalence of 10.4% was found using ICD-10 codes, a method that may overlook the undiagnosed cases and, therefore, underestimate the exact prevalence [30]. The prevalence calculated based on DASS in surgery candidates was 22.8% [32], while the highest prevalence (of 38.6%) was identified via HADS [41]. Song et al. [30] found no differences in terms of pre-existing (*p* = 0.601) or newly onset anxiety (*p* = 0.089) between operative and non-operative subjects [30]. Compared to controls, some data suggested higher rates of anxiety in PHPT [33], while others did not [41]. In a large study on patients with hypercalcemia, patients at high risk of PHPT diagnosis had higher anxiety rates than matched controls (15.4% versus 10.8%, *p* < 0.001), but the study did not compare the subjects confirmed with PHPT diagnosis versus controls [33]. Liu et al. [38] found higher anxiety scores in individuals who underwent parathyroidectomy compared to those who were referred for thyroidectomy (4.5 ± 5.6 versus 3.2 ± 4.1, *p* < 0.01) [38]. On the other hand, Kunert et al. [41] found a similar anxiety prevalence in people with PHPT who had to undergo parathyroidectomy (38.6% versus 34%, *p* > 0.05) [41]. Of note, the controls had non-toxic goitre, an affliction that may have higher rates of anxiety than the general population according to some authors [43].

The severity of anxiety was analysed in 101 surgery candidates with PHPT. Most patients had mild (9.9%) or moderate (7.9%) anxiety, while 4% had severe anxiety. Extremely severe anxiety was identified in 1% of patients [32]. Other studies only provided anxiety scores, not the prevalence. For instance, Chan et al. [28] reported a mean score of 5.17 (1.01), corresponding for a lack of anxiety [28]. Similarly, Koman et al. [37] reported a median score of 6 (4–10) in patients with PHPT ≥ 50 years [37]. Wang et al. [40], however, found a higher score of proneness to anxiety (37.39 ± 10.34), which indicated a moderate tendency towards anxiety, according to the State–Trait Anxiety Inventory. The mean score (of 35.43 ± 11.56) was indicative for low anxiety [40] (Table A7).

#### 3.2.2. Analysis of the Anxiolytic Medication

Data regarding medication against anxiety in patients confirmed with PHPT were provided in a single study that reported a prevalence of 13.3%, surgery candidates being at 1.4-times higher risk of anxiolytic drug use compared to a matched control population [OR = 1.4 (1.31–1.5)] [34].

#### 3.2.3. Endocrine Assays and Anxiety

Limited data (n = 5 studies) suggested that patients with PHPT might be more affected by anxiety at lower calcium levels [30,34,38,40,41]. For instance, a negative weak correlation between ionized serum calcium and anxiety severity was found (r = −0.1863, *p* < 0.05) [41]. The highest rate of using anxiolytics in PHPT versus controls was confirmed at lower calcium levels (*p* = 0.004) by Koman et al. [34]. On the other hand, a retrospective study in 192 patients with PHPT found no correlation between the current anxiety (r = −0.060, *p* = 0.726) or anxiety proneness (r = 0.049, *p* = 0.782) and serum calcium levels [40]. A study on 244 subjects with PHPT also did not confirm this type of correlation [38]. While one study reported a small positive correlation between anxiety and serum PTH levels (r = 0.1797, *p* < 0.05) [41], others did not [38,40], nor to a potential correlation with serum cortisol [40]. Blood osteocalcin was negatively associated with current anxiety after multiple variable adjustments (r = −0.426, *p* = 0.027) in a single study [40] (Table A8).

#### 3.2.4. The Potential Parathyroidectomy Impact on Anxiety

Most studies showed a reduction in anxiety scores following parathyroidectomy [28,32,34,38]. The largest of them (N = 244 patients who underwent parathyroidectomy for PHPT) reported lower anxiety scores postoperatively [mean difference of 2.5 (1.9–3.1), *p* < 0.01]. The same study compared parathyroidectomy with thyroidectomy candidates and found that, although preoperative scores were higher in subjects with PHPT (4.5 ± 5.6 versus 3.2 ± 4.1, *p* < 0.01), following surgery, anxiety scores were similar to those in persons who underwent thyroidectomy (2.0 ± 3.6 versus 2.1 ± 3.9, *p* = 0.75). Moreover, after parathyroid surgery, anxiety severity improved (moderate to severe: 18% versus 5.3%, *p* < 0.01) [38]. Another study (N = 101 patients with PHPT who were surgically treated) identified a reduction in anxiety scores, as assessed by DASS, of 62.4%. The lowering of anxiety scores was statistically significant, both at one-month post-surgery (*p* = 0.001) and at six months (*p* = 0.001). Similar results were reported via SCL90R [32]. Lower anxiety scores were also reported in a small study in 36 surgery patients, both at one week and at three months following surgery [5.17 (1.01) versus 3 (1.01) versus 2.46 (1.03), *p* = 0.01]. The trend, however, was not statistically significant (*p* = 0.2) [28]. Regarding anxiolytic medication, one large study (N = 8279 patients with PHPT) identified a higher risk following surgery compared to controls [0–6 months: RR (CI) = 1.31 (1.2–1.43)] but decreased over time (*p* for trend = 0.003). Prevalent users, however, had similar risk to controls [0–6 months: RR (CI) = 0.97 (0.9–1.05)] [34]. As mentioned, Scerrino et al. [35] compared cholecystectomy candidates and found that individuals who underwent parathyroidectomy had similar anxiety rates preoperatively (72% versus 93%, *p* = 0.7321), at six months (76.7% versus 97.6%, *p* = 0.3823) and at one year (83.72% versus 97.6%, *p* = 0.4329) after surgery, respectively [35] (Table A9).

#### 3.2.5. Cinacalcet Use Before Parathyroidectomy and Anxiety Impact

One prospective observational study analysed the impact of preoperative exposure to cinacalcet in adults with PHPT; similar anxiety scores were found after four weeks into the drug compared to baseline (*p* = 0.140). Following parathyroidectomy, however, the median anxiety scores decreased [−2 (0, −3), *p* = 0.040] [37] (Table A9).

#### 3.2.6. Conservative Management Versus Parathyroidectomy in Patients with PHPT and the Consequences on Anxiety

Song et al. [30] investigated the differences regarding the prevalence and incidence of anxiety in PHPT with concern to conservative management versus parathyroidectomy. The study reported no statistically significant difference in terms of pre-existing anxiety (10.8% versus 10%, *p* = 0.601) and newly onset anxiety (14.3% versus 11.6%, *p* = 0.089) [30] (Table A9).

## 4. Discussion

The studies (n = 16, N = 10,325) that we identified based on the mentioned methods showed an interesting association between depression and/or anxiety and the features of PHPT, either in terms of diagnosis or management [27,28,29,30,31,32,33,34,35,36,37,38,39,40,41,42], but the data are not homogenous (Figure 2).

### 4.1. Tools and Methods Used for the Assessment of Depression and Anxiety

One of the most used depression scales in recent studies is BDI [44] and its revised version, BDI-II [29,31,32,36,40,41]. The first version underwent multiple revisions, and the latest version, BDI-II, was adjusted to align with DSM-IV criteria for depressive disorder [45]. Even though BDI-I and BDI-II have been validated and are extensively used in different medical areas [46,47,48,49,50], they might overestimate the prevalence of depression given that factors, such as fatigability, sleep disturbances, osteoporosis, fractures, weight loss and loss of appetite, commonly found in various chronic illnesses, are included [51,52,53]. Of note, fatigability, bone and muscle ailments, as well as sleep disturbances are highly prevalent in subjects with PHPT, and symptoms such as loss of appetite and weight loss may be related to nausea and vomiting due to hypercalcemia [54,55]. Liu et al. [38] applied PHQ-9, which evaluates both the presence of depression and its severity. However, it takes into consideration multiple symptoms, such as sleep troubles, tiredness, and appetite changes, that might not be a reflection of PTH and serum calcium excess [56,57]. Another scale that has been validated among different populations is DASS [52] [58,59,60]. Unlike BDI and PHQ-9, DASS does not include physical symptoms like fatigue and appetite changes, and it also differentiates between anxiety, stress, and depression [52]. HADS served as a tool in PHPT investigation amid two recent studies [37,41]. This widely validated scale might be more fitting as it avoids somatic symptoms and does not overlap with common symptoms of (non-PTH-related) chronic illness [61,62]. However, recently, it has been suggested that cut-off values should be adjusted in certain patients [63].

The anxiety scales that have been used in patients with PHPT in these recent studies included GAD-7 [35,38], HADS [37,41], DASS [28,32], and State–Trait Anxiety Inventory [40]. All are widely used and validated scales, and they mostly focus on subjective symptoms [64]. The main advantages of scales like HADS or DASS are the ability to evaluate patients in a time-efficient manner and with little prior training [65]. One of the major limitations of applying these scales in assessing depression and anxiety is the fact that they are non-diagnostic screening tools [66]. Another important point is the method of administration, considering that self-rating was shown to provide higher scores compared to interview administration, especially in the elderly [67]. There were also studies based on ICD-10 codes [30,34]. Although these may include an objective diagnosis, they may not always be accurate, as some authors suggested an underestimation of depression prevalence [68]. Overall, while data from questionnaires are practical, they might be biased due to relying on self-reporting symptoms; yet, they may be integrated in the evaluation of one patient with PHPT as a screening tool in daily endocrine practice. Further studies correlating standardized objective clinical assessments are needed to understand the extent of the PHPT effect on depression and anxiety.

### 4.2. Common Pathogenic Mechanisms Underlying PHPT and Depression/Anxiety

The mechanisms linking PHPT and psychiatric manifestations such as depression and anxiety are intricate and incompletely understood so far [69]. The influence of the parathyroid hormone on the brain is supported by the presence of PTH receptors, namely, PTH1R and PTH2R, in various brain regions. However, its ability to cross the blood–brain barrier is disputed, with no clear data [70]. The PTH1R receptor found in brain tissue is expressed in distinct regions of the limbic system, including the thalamus, amygdala, and entorhinal cortex, in vestibular nuclei, in the cerebellum, and astrocytes, being identical to the PTH1R from bones and kidneys according to animal studies [71]. Both PTH and PTHrP can bind to PTH1R. PTH1R activation was shown to reduce neuro-inflammation, to improve cell proliferation, and increase the blood flow in the brain [72,73]. Another PTHR that was found in the brain was PTH2R, which was described in 1995 by Usdin et al. [74]. Unlike PTH1R, PTH2R differentiates between PTH and PTHrP, and it is only activated by the former [75]. While PTH can bind to PTH2R, a much more potent ligand is TIP39. Not only does PTH2R have a higher affinity for TIP39 but the response is also more sustained [69]. The binding of TIP39 to PTH2R was linked in animal studies to fear, nociception, reproductive behaviour, and stress response by modulating the hypothalamic–pituitary–adrenal axis [76]. It was also shown to reduce depression and anxiety-like behaviours [76,77,78].

Apart from the possible effects of PTH on the central nervous system, other mechanisms involved in the development of depression and anxiety in patients with PHPT might be related to the influence exerted by excessive PTH and calcium on monoamine neurotransmitters; however, data are scarce and conflicting to date [79]. While it has been proposed that hypercalcemia might increase the neurotransmitters by reducing monoamine oxidases (MAO) activity and reducing the catabolism of serotonin, dopamine, and norepinephrine [79], other studies suggested quite the opposite, namely, that intracellular calcium stimulates MAO-A [80]. Overall, the mechanisms connecting PHPT to the development of depression/anxiety are complex and involve both the direct and indirect effects of PTH and calcium. Further studies are needed for a more in-depth understanding of these pathogenic pathways.

### 4.3. Confounding Factors in Assessing Anxiety/Depression in Patients with Parathyroid Tumours

#### 4.3.1. Age and Gender

In order to accurately understand the relationship between PHPT and depression/anxiety, possible confounding elements must be taken into account. First of all, PHPT has a peak of incidence in menopausal women [81]. This is an important factor when assessing depression in patients with PHPT, as it has been shown that women are 2- to 2.5-times more likely to develop depression and major depressive disorder than men, with postmenopausal women having the highest risk [82,83]. Recent studies have shown that the depression affects about one-third of menopausal women worldwide, and it is influenced by various factors, including marital, social, and economic status, as well as the presence of menopausal symptoms [84,85]. Not only are women more likely to develop depression but anxiety might also be more frequent in this population due to various psychosocial and biological contributors [86,87]. Depression may have an age-related incidence, as recent studies have found peaks in middle adulthood and seniors [88,89]. In order to rule out this bias, some studies compared patients with PHPT with age- and sex-matched controls and found higher depression scores in patients with PHPT [29,31], suggesting that PHPT itself leads to a greater depression rate. While depression scores decreased following parathyroidectomy, they may remain higher than in controls [29]. Interestingly, the number of livebirths was shown to be a protective factor against depression in women [90], but Febrero et al. [31] found that in patients with PHPT, having children was associated with higher depression scores [31].

#### 4.3.2. Comorbidities

The relationship between PHPT, comorbidities such as metabolic syndrome and heart disease, and depression and/or anxiety is intricate and often multidirectional. PHPT might be accompanied by multiple ailments, ranging from classical ones such as osteoporosis/osteoporotic fractures and chronic kidney disease to cardiovascular disease, type 2 diabetes mellitus, and metabolic syndrome. One of the more recently characterized complications of PHPT is diabetes, which not only has a higher prevalence in patients with PHPT but, also, displays a positive linear association with the value of serum calcium [91]. Depression and/or anxiety are also linked to diabetes in a bi-directional manner. While there is a causal relationship from diabetes to depression/anxiety, they also increase the risk of developing diabetes and are linked to poorer glycaemic control and, consequently, a higher disease burden and mortality [92,93,94].

Regarding metabolic syndrome, both depression and PHPT are contributing factors, and their independent and cumulative effect ought to be investigated [95,96,97]. Furthermore, metabolic syndrome might increase the risk of anxiety via chronic inflammation [98]. One recent study that we already mentioned [32] briefly covered the complex relationship between mental health, PHPT, and type 2 diabetes mellitus. The authors found that in subjects confirmed with PHPT, there were no statistically significant differences regarding depression and anxiety scores, as evaluated using multiple scales, between diabetic and non-diabetic individuals following parathyroid surgery [32]. Little is known of the cumulative effect of PHPT and depression and/or anxiety on diabetes. Further studies are needed to understand this complex relationship.

Apart from the link with PHPT, osteoporosis has been connected to depression both as a possible risk factor and as a possible effect. On the one hand, osteoporosis has been shown to independently increase the risk of depression [99]. Moreover, fragility fractures, orthopaedic surgery, and long-term immobilization may trigger depression [100,101]. On the other hand, depression has previously been associated with an increased risk of osteoporosis [102]. Multiple common mechanisms have been proposed, including the chronic inflammatory state, leading to apoptosis of osteoclasts, and the disruption of the endocrine system such as the decrease in oestrogens and testosterone and the increase in cortisol [103]. Another important aspect is the fact that drugs acting against osteoporosis might have depression as adverse reactions. For instance, when compared to teriparatide, alendronate was associated with a higher risk of depression [patients ≤65 years: ROR (reporting odds ratio) of 14.67 (11.55–18.63), individuals >65 years: 3.60 (2.82–4.59)] and anxiety [subjects ≤65 years: ROR of 7.10 (5.79–8.71), individuals >65 years: 2.28 (1.84–2.84)] in one study [104].

Depression risk in PHPT may also be mediated by the presence of cardiovascular involvement. In turn, depression and PHPT may exert a cumulative role on the development of cardiovascular conditions. Although not an indication for surgery, increased cardiovascular risks due to arterial stiffness, endothelial dysfunction, and high-risk hypertension are associated with PHPT [105,106]. Cardiovascular disease is also associated with an elevated risk of depression and/or anxiety [107]. On the other hand, depression and/or anxiety might contribute to the development of a cardiovascular disorder by speeding up the development of risk factors and promoting arterial stiffness [108,109,110].

#### 4.3.3. Different PHPT Disease Forms: From Normocalcemic to Hypercalcemic Type

In recent years, the presentation of the PHPT patient changed, with disease being diagnosed in early stages, most frequently in asymptomatic stages, and even in patients with normocalcemia [111,112]. The various forms of PHPT raised the question whether depression and/or anxiety have different characteristics in asymptomatic and normocalcemic patients. Patients with PHPT are considered asymptomatic if they lack classical symptoms (e.g., osteoporosis, fragility fractures, kidney stones, and severe neurological consequences of the acute hypercalcemia) [113]. However, asymptomatic patients may present with various other minor complaints and, more often, have been shown to have different types of psychological symptoms when compared to the healthy population [114]. Moreover, the severity of the symptoms in PHPT is not always reflected by the PTH and calcium levels [115]. Regarding asymptomatic PHPT, Jovanovic et al. [32] showed that depression and anxiety had a high prevalence, and both comorbidities improved following parathyroidectomy [32]. Vadhwana et al. [39], however, did not find a reduction in anxiety/depression scores in subjects with asymptomatic PHPT, even though patients with symptomatic disease had better scores following surgery [39]. Normocalcemic PHPT has been recently recognized as a distinct disease form, characterized by elevated PTH levels and normal serum calcium in the absence of secondary causes for a PTH increase [116,117]. Older data, like those from a case report of an 80-year-old male with depression resistant to treatment, who experienced resolution of depression during a post-parathyroidectomy follow-up of three years, suggested that there might be a possible connection between normocalcemic PHPT and depression [118]. As mentioned, we did not identify distinct studies regarding normocalcemic patients and depression/anxiety. Notably, only a cohort from 2024 studied by Febrero et al. [29] included normocalcemic patients among hypercalcaemic subjects, but the study did not compare the two types of PHPT [29]. Given the limited data available, further studies are needed in order to determine the impact of the PHPT phenotype on depression and/or anxiety.

### 4.4. Current Limits and Further Perspectives

As limits of the current work, we mention that this was a non-systematic review since we did not intend to restrain the included studies, noting that their design and methods of depression/anxiety assessments varied. Also, we only included cohorts that specifically provided data with respect to depression and/or anxiety, which might remain undiagnosed in other studies that analysed other elements of the PHPT-related phenotype. Other related topics are still at a low level of statistical evidence, such as the impact of medical therapy against hypercalcemia on depression/anxiety versus surgery, the extended panel of psychiatric manifestations in PHPT (other than depression and anxiety), or the impact of the disease duration (PHPT or depression/anxiety) on the clinical picture. We found only limited data on cinacalcet, as mentioned. Generally, the normalization of serum calcium is crucial in hypercalcemia-induced neuro-psychiatric symptoms, as seen in the setting of PHPT and apart from the surgical approach, and, hence, cinacalcet may be used for lowering blood calcium [119]. Previous reports have shown that it may manage psychotic symptoms, too, but these remain at the case report level [120]. The question arises whether cinacalcet truly impacts depression and/or anxiety. As mentioned, a single study included patients treated with cinacalcet, and the drug was shown to reduce depression scores but did not impact anxiety scores [37]. Depression is not only a common comorbidity in patients with PHPT but it may sometimes be one of the first manifestations. For example, a recent paper reported the case of a 72-year-old male who was diagnosed with PHPT due to an ectopic mediastinal parathyroid adenoma after presenting with fatigue and depression, both of which improved and then remitted following the surgical cure [121]. Another case report suggested that newly onset depression and cognitive decline might be accompanying symptoms of PHPT, especially in the elderly. Vultaggio et al. [122] reported that an 80-year-old patient who presented with a soporous state, depression, and cognitive decline was diagnosed with a parathyroid carcinoma and had a neurological recovery after parathyroidectomy [122]. Other psychiatric symptoms may also raise the alarm for an underlying parathyroid pathology, such as PHPT (e.g., delirium and psychosis) [123,124]. While a surgical cure leads to the resolution of these symptoms in some cases [125,126], other patients with long-standing hypercalcemia are unresponsive to this treatment, in spite of the surgical cure for the parathyroid tumour [127,128]. Even though high levels of calcium are attributed to involvement in the onset of psychosis, in some cases, even mild hypercalcemia might induce psychosis [129]. The psychiatric pathology of patients with PHPT might be complex, with more manifestations overlapping [130]. These data support the idea that depression, anxiety, fatigue, and even psychosis are important symptoms of PHPT and may lead to an early diagnosis in patients otherwise asymptomatic, and further trials are necessary to pinpoint the prevalence and severity of the pathogenic connection with high PTH and/or calcium, in addition to finding more evidence for which sub-group of patients is at a higher risk for such ailments and if PHPT management should be adjusted under these circumstances.

## 5. Conclusions

The modern approach for patients with PHPT should be complex and go beyond classical symptoms. Recent studies revealed that mental health conditions such as depression and anxiety have a high prevalence in patients with PHPT, regardless of the traditional symptoms, and are associated with increased medication use. Moreover, the scores of depression were generally higher compared to control populations. However, the underlying mechanisms remain incompletely elucidated. No correlations between depression or anxiety and serum calcium levels were found, while PTH had a slight positive correlation with depression. Parathyroidectomy appears to be beneficial for these mental health aspects in subjects with PHPT as it improves the scores, prevalence, and severity of depression and reduces anxiety scores. Improvements in symptoms following surgery in otherwise asymptomatic patients raise the question whether depression might be taken into consideration when referring patients for surgery. Cinacalcet might reduce depression scores, although more evidence is needed. Women are at higher risk both for PHPT and depression and anxiety. The optimal method for depression and anxiety screening for PHPT patients remains to be determined, and the current scales need validation and perhaps adjustment for this specific population group.

## Figures and Tables

**Figure 1 diseases-13-00054-f001:**
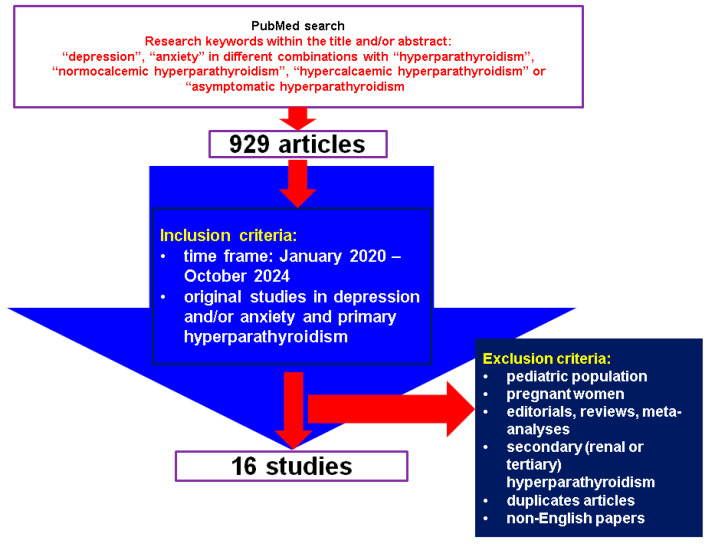
Flow chart diagram of search.

**Figure 2 diseases-13-00054-f002:**
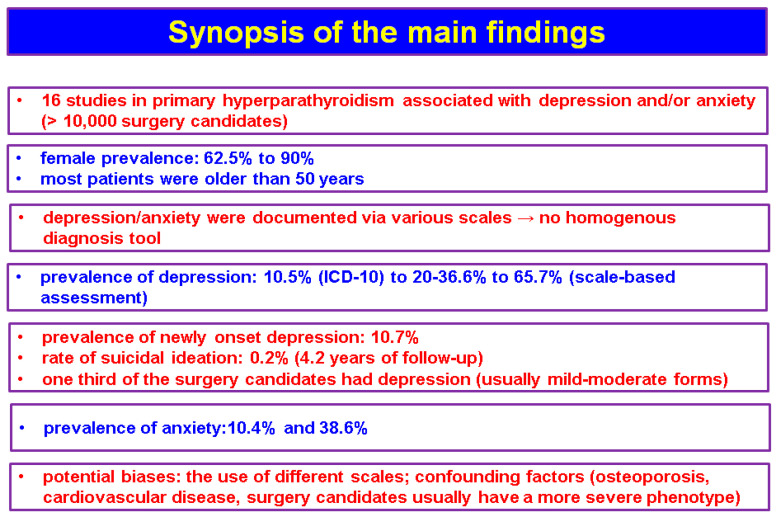
Synopsis of the main findings according to our methods.

**Table 1 diseases-13-00054-t001:** Studies investigating depression and anxiety in patients with PHPT (the display starts with the most recent publication date) [27,28,29,30,31,32,33,34,35,36,37,38,39,40,41,42].

First Author/Year of Publication/ Reference	Study Design/Studied Population	Diagnostic Method (Tool) for Depression	Diagnostic Method (Tool) for Anxiety
Bunch 2024 [27]	Retrospective study N = 17,491 with hypercalcemia F:M = 10,924:6567 (62.5% females) Mean age = 59 ± 17.3 y N1 = 6096 who underwent screening for PHPT F:M = 4171:1925 (68.4% females) Mean age = 62.8 ± 14.6 y N2 = 11,395 without screening for PHPT F:M = 6753:4642 (59.3% females) Mean age = 56.9 ± 18.2 y	ICD-10 codes	NA
Chan 2024 [28]	Longitudinal prospective study N = 36 with PHPT who underwent PTX F:M = 33:3 (92% females) Mean age = 59 ± 12.8 y	Depression Anxiety Stress Scales	Depression Anxiety Stress Scales
Febrero 2024 [29]	Prospective case-control N1 = 49 with PHPT (hypercalcemic and normocalcemic) who underwent PTX F:M = 39:10 (80% females) Mean age = 63.78 ± 10.09 y N2 = sex- and age-matched-controls	Beck Depression Inventory-II	NA
Song 2024 [30]	Retrospective cohort study N = 3728 with PHPT After propensity score match: N1 = 959 non-operative patients F:M = 766:193 (79.9% females) Mean age = 62 ± 14 y N2 = 959 operative patients F:M = 763:196 (79.6% females) Mean age = 62 ± 13 y	ICD-10 codes	ICD-10 code
Febrero 2023 [31]	Prospective cohort N1 = 65 with PHPT (hypercalcemic and normocalcemic) F:M = 50:15 (77% females) Mean age: <40 y: 11 (17%) 40–60 y: 25 (38%) >60 y: 29 (45%) N2 = 65 sex and age matched-controls	Beck Depression Inventory-II	NA
Jovanovic 2023 [32]	Prospective study N = 101 with asymptomatic PHPT who underwent PTX F:M = 88:13 (87.1% females) Average age = 60.7 (range 27–80) y	Beck Depression Inventory Depression Anxiety Stress Scales Symptom Check List 90-revised	Depression Anxiety Stress Scales Symptom Check List 90-revised
Lorenz 2022 [33]	Retrospective cohort study N = 135,034 with hypercalcemia F:M = 96,554:38,466 (72% females) Mean age = 63 ± 10 y N1 = 13,136 with PHPT diagnosis N2 = 20,176 with high risk of PHPT diagnosis with PTH data N3 = 24,905 with high risk of PHPT diagnosis without PTH data	patient records	patient records
Koman 2022 [34]	Retrospective case-control and prospective cohort analyses N1 = 8279 who underwent PTX F:M = 6374:1905 (77% females) N2 = 82,790 matched controls F:M = 63,740:19,050 (77% females) Age: <30 y = 2.2%, 30–39 y = 4.1%, 40–49 y = 11.4%, 50–59 y = 22%, 60–69 y = 29.6%, 70–79 y = 23.5%, 80+ y = 7.2%	ICD-10 (F06–F99)	ICD-10 (F06–F99)
Scerrino 2022 [35]	Retrospective study N1 = 43 with PHPT who underwent PTX F:M = 33:10 (76.7% females) Mean age = 52.4 y N2 = 233 who underwent thyroidectomy F:M = 185:48 (79.4% females) Mean age = 54.6 y N3 = 43 who underwent cholecystectomy F:M = 34:9 (79% females) Mean age = 52.4 y	Hamilton Depression Rating Scale	Generalized Anxiety Disorder-7
Szalat 2022 [36]	Prospective case-control N = 18 with PHPT who underwent PTX F:M = 14:4 (79% females) Mean age = 67.9 ± 7.6 y	Beck Depression Inventory	NA
Koman 2021 [37]	Prospective observational N = 35, with age ≥ 50 y (out of which N1 = 19 with age ≥ 70 y) with PHPT and mild cognitive impairment who underwent cinacalcet treatment 4 weeks before PTX F:M = 31:4 (88.6% females) Median (IQR) age = 71 (62–79) y N1: F:M = 17:2 (89.5% females) Median (IQR) age = 77 (72–82) y	Hospital Depression and Anxiety Scale	Hospital Depression and Anxiety Scale
Liu 2021 [38]	Prospective multi-centric observational study N = 405 candidates to endocrine surgery F:M= (76.6% females) Mean age = 59 ± 13.9 y N1 = 244 who underwent PTX F:M = 192:52 (78.7% females) Mean age = 63 ± 12.2 y N2 = 161 who underwent thyroidectomy F:M = 120:41 (74.5% females) Mean age = 52.4 ± 14.2 y	Patient Health Questionnaire-9	Generalized Anxiety Disorder-7
Vadhwana 2021 [39]	Prospective study N = 78 with PHPT who underwent PTX F:M = 56:22 (72% females) Median (IQR) age = 62 (52–70) y N1 = 28 asymptomatic PHPT F:M = 20:8 (71% females) Median (IQR) age = 69 (58–73) y N2 = 50 symptomatic PHPT F:M = 36:14 (72% females) Median (IQR) age = 58 (50–70) y	EuroQOL-5D-3L	EuroQOL-5D-3L
Wang 2021 [40]	Retrospective study N = 192 with PHPT F:M = 147:45 (76.6% females) Mean age = 52.7 ± 13.8 y	Beck Depression Inventory	State-Trait Anxiety Inventory
Kunert 2020 [41]	Observational retrospective cohort study N1 = 101 with PHPT who underwent PTX F:M = 85:16 (84.15% females) Median (range) age: 60 (20–86) N2 = 50 controls F:M = 42:8 (84% females) y Median (range) age: 58 (25–75) y	Hamilton Depression Rating Scale Beck Depression Inventory—II Hospital Anxiety and Depression Scale	Hospital Anxiety and Depression Scale
Weber 2020 [42]	Retrospective N = 125 with PHPT who underwent surgery F:M = 95:30 (76% females) Median age = 60.4 (23–83) y	Patients’ records	NA

Abbreviations: F = females; ICD-10 = The International Classification of Diseases-Tenth Revision Clinical Modification; M = males; N = number of patients; NA = not available; PHPT = primary hyperparathyroidism; PTX = parathyroidectomy; y = years; blue font = study design; red font = studied population.

## Data Availability

Not applicable.

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
