# Peer review of "An Analysis of Primary Hyperparathyroidism in Association with Depression or Anxiety"

_diseases, 2025, doi:10.3390/diseases13020054_

Round 1
Reviewer 1 Report
Comments and Suggestions for Authors
Primary Hyperparathyroidism (pHPT) is a common disease especially in elderly women, and mainly is detected be routine exmainations by blood test.
It is long known that pHPT is assiated with different psychological and/or psychiatric dieseases,.
This is a comprehensive review of all publications published in English language within the last 4 years, so a very short time. Mainly the prevalence of depression in those patients should be elaborated. There are, however, several problems, most of those are already discussed by the authors
1. The main critical point is the right diagnosis of depression, and the questionairs that had been used.
2. Nothing known about the duration of both pHPT as well as the psychiatric problems
3. It is not elabotated, whether the height of iodized calcium, or at least albumin corrected calcium plays a role, mainly of cause because this is not mentioned in the manuscrpit.
4. Most of the studies are retrospective or crossectional, only one, low number but prospective and controlled, and clear results. Only those studies are worthwhile to do and it should be emphasized to initiating more such studies.
5. It is very difficult to read and understand the message, because of the different results, that are confusing, mainly because of the low quality- retrspective, crossectional, without out clear dividing between serious or only borderline high cvalcium and without mention albumin corrected of ionized calcium levels
6. The manuscript is far to long - takes a lot of time to read and understand- and the message unfortunately is confuse.
7. It had been a lot of work, done by the authors, but if they are not from beginning on select only sound manuscripts, the final message has to be unclear. It would habe been better to select prospective controlled studies within a longer period of time and more studies could be included and uncontrolled excluded
8. We would recommend to put all the tables except Tab. 1 in Appendix, and try to shorten the manuscript to about 50% length.
Author Response
Response to Review 1 Comments
Dear Reviewer,
Thank you very much for your time and your effort to review our manuscript.
We are very grateful for providing your valuable feedback on the article.
Here is our response and related amendment that has been made in the manuscript according to your review (marked in yellow color).
Primary Hyperparathyroidism (pHPT) is a common disease especially in elderly women, and mainly is detected be routine examinations by blood test. It is long known that pHPT is associated with different psychological and/or psychiatric diseases. This is a comprehensive review of all publications published in English language within the last 4 years, so a very short time.
Thank you very much. We intended to present the most recent data amid modern medicine in the field of primary hyperparathyroidism considering the modern topic of the mental health and its potential connection to the mentioned anomaly of the mineral metabolism. Indeed, the most recent data involves a 5-year frame which has been provided since the objective was an up-date in this matter considering the shifting of the presentation by early detection amid calcium screening protocols and easy access to blood assays. Thank you
Mainly the prevalence of depression in those patients should be elaborated.
Thank you very much. This aspect has been largely approached in the section 3.1.1. and associated tables. Thank you.
There are, however, several problems; most of those are already discussed by the authors.
Thank you very much. We addressed them point by point as follows:
The main critical point is the right diagnosis of depression, and the questionnaires that had been used.
Thank you very much. We addressed this issue in the section 4.1. This reflects and involves a general problem/concern that is still a matter of debate – which is the most adequate tool to assess depression or anxiety in these distinct endocrine patients amid every day practice (as a routine, not experimental clinical aspect). We elaborated a more complex discussion/insight in the mentioned section. Thank you
Nothing known about the duration of both pHPT as well as the psychiatric problems.
Thank you very much. According to your recommendation, this aspect was mentioned at Limitation section: “Other related topics are still at low level of statistical evidence such as ….the impact of the disease duration (PHPT or depression/anxiety) on the clinical picture.”
Thank you
It is not elaborated, whether the height of iodized calcium, or at least albumin corrected calcium plays a role, mainly of cause because this is not mentioned in the manuscript.
Thank you very much. The impact of the endocrine/biochemistry assays regarding the confirmation of the primary hyperparathyroidism was analyzed for each sub-section with respect to depression/anxiety outcomes. For example, we mentioned:
“Limited data (n=5 studies) suggested that patients with PHPT might be more affected by anxiety at lower calcium levels [30,34,38,40,41]. For instance, a negative weak correlation between ionized serum calcium and anxiety severity was found (r=-0.1863, P<0.05) [41]. The highest rate of using anxiolytics in PHPT versus controls was confirmed at lower calcium levels (P=0.004) by Koman et al. [34]. On the other hand, a retrospective study in 192 patients with PHPT found no correlation between the current anxiety (r=-0.060, P=0.726) or anxiety proneness (r=0.049, P=0.782) and serum calcium levels [40]. Nor a study in 244 subjects with PHPT confirmed this type of correlation [38]. While one study reported a small positive correlation between anxiety and serum PTH levels (r=0.1797, P<0.05) [41], others did not [38,40], neither with concern to a potential correlation with serum cortisol [40]. Blood osteocalcin negatively associated with current anxiety after multiple variables adjustments (r=-0.426, P=0.027) in a single study [40]. (Table S9)”
Also, the analysis based on the data displayed in Table 4 (S4) reflects the values of the biochemical parameters such as serum calcium panel.
e.g. sneak peek into table S4.
Thank you.
Most of the studies are retrospective or cross-sectional, only one, low number but prospective and controlled, and clear results. Only those studies are worthwhile to do and it should be emphasized to initiating more such studies.
Thank you very much. This is the current level of evidence at this point and we specified the study design for each cited paper. We mentioned that depression and anxiety in patients with primary hyperparathyroidism is not included in current guidelines for surgery candidates, hence, the topic is still open. Thank you
It is very difficult to read and understand the message, because of the different results, that are confusing, mainly because of the low quality- retrospective, cross-sectional, without out clear dividing between serious or only borderline high calcium and without mention albumin corrected of ionized calcium levels.
Thank you very much. This is an analysis of prior published papers. According to your recommendation, we revised the presented data, reduced the length of the article and moved all the tables as Adnexa (except for Table 1). As mentioned, all the available data with concern to the calcium assays has been provided, as seen in the area of different study designs. Thank you.
The manuscript is far too long - takes a lot of time to read and understand- and the message unfortunately is confuse.
Thank you very much. This issue has already been addressed below. Thank you
It had been a lot of work, done by the authors, but if they are not from beginning on select only sound manuscripts, the final message has to be unclear. It would have been better to select prospective controlled studies within a longer period of time and more studies could be included and uncontrolled excluded.
Thank you very much. To our best aware, there are no other prospective controlled studies in this specific matter according to our methods, that is why it is crucial to address this topic from a multidisciplinary perspective and to offer a complex presentation/analysis. Thank you
- We would recommend putting all the tables except Tab. 1 in Appendix, and try to shorten the manuscript to about 50% length.
Thank you very much. According to your recommendation, we revised the data, reduced the length of the article and moved all the tables as Adnexa (except Table 1).
Of note, to our best knowledge, this is the most complex analysis that we identified in the field of primary hyperparathyroidism and its association with depression or anxiety. Also, this is the result of a massive team work in order to identify the complex issues, pitfalls, and challenges in the mentioned area. Moreover, this is a double perspective: the one of the specialist that usually treats patients with endocrine (parathyroid) tumors (involving the endocrinology and the endocrine surgery departments), but, also, from the point of view of the practitioners that might involve patients confirmed with depression or anxiety, including primary health care. Hence, the complexity of the work involves a certain length of the paper. Notably, this is no length restriction according to MDPI rules, but a mandatory minimum number of words for reviews. Thank you
Thank you very much.

Reviewer 2 Report
Comments and Suggestions for Authors
The authors do a good job in their organization and presentation of the review on hyperparathyroidism in association with depression or anxiety. However, there are a few methodological considerations and corresponding results that need to be considered.
It was not stated if this was meant to be "systematic review". If it is a systematic review, the authors should refer to PROSPERO, which is where systematic reviews are registered for most MDPI journals.
Even if the authors are not declaring this literature review as "systematic", it still needs an overhaul in expressing its methodology and results.
First, the methods are extremely sparse. The authors did not include the databases searched or time span of publication (years). Second, the authors did not include sufficient inclusion and exclusion criteria. For a study of this nature - whether it technically a systematic review or not - the authors should include a PRISMA diagram or a similar flowchart diagram to illustrate the inclusion and exclusion of studies, which is necessary to support why these 16 were chosen. The authors should include the initial number of papers identified with search criteria and the number(s) that were included for each reason in the exclusion criteria.
The authors did a good job of extracting relevant quantitative information. However, the authors really missed an opportunity to aggregate that data to present into summary statistics - e.g. a form of metadata analysis (which is different from meta-analysis, which would include a meta regression equation). For example, the co-presence of depression or anxiety could have been calculated using aggregated data and presenting an odds ratio or risk ratio. Another option that is often done to aggregate diverse data in reviews such as this is to measure the association(s) of interest by calculating a Cohen's D.
The authors need to improve the display items. Right now they have very long and difficult to navigate tables. Inclusion of longer tables could be done using the Appendix, which would still allow it to be included with the main paper file download the way MDPI does their formatting. But, more importantly, the authors need to devise a better way to summarize the information using either a more compact table, or preferably, with the use of a figure (pie chart, bart chart, etc.).
Author Response
Response to Review 2 Comments
Dear Reviewer,
Thank you very much for your time and your effort to review our manuscript.
We are very grateful for your insightful comments and observations, also, for providing your valuable feedback on the article.
Here is a point-by-point response and related amendments that have been made in the manuscript according to your review (marked in yellow color).
The authors do a good job in their organization and presentation of the review on hyperparathyroidism in association with depression or anxiety.
Thank you very much. We really appreciate it!
However, there are a few methodological considerations and corresponding results that need to be considered.
Thank you very much. We addressed them point by point.
It was not stated if this was meant to be "systematic review". If it is a systematic review, the authors should refer to PROSPERO, which is where systematic reviews are registered for most MDPI journals. Even if the authors are not declaring this literature review as "systematic", it still needs an overhaul in expressing its methodology and results.
Thank you very much. We specified in section 4.4. the fact that the current analysis is not a systematic review due to the current level of statistical evidence that involves studies with different designs and we intended to cover a wider area of issues. However, non-systematic reviews are well respected articles that allow a more flexible approach in the domains that are still a matter of debate. Thank you
First, the methods are extremely sparse. The authors did not include the databases searched or time span of publication (years). Second, the authors did not include sufficient inclusion and exclusion criteria. For a study of this nature - whether it technically a systematic review or not - the authors should include a PRISMA diagram or a similar flowchart diagram to illustrate the inclusion and exclusion of studies, which is necessary to support why these 16 were chosen. The authors should include the initial number of papers identified with search criteria and the number(s) that were included for each reason in the exclusion criteria.
Thank you very much. We included the flowchart, time frame, data base, search terms, and inclusion/exclusion criteria. PRISMA stands for “Preferred Reporting Items for Systematic Reviews and Meta-Analyses” (http://prisma-statement.org/prismastatement/flowdiagram.aspx) and this is a non-systematic review. Due to the heterogeneity of the spectrum in primary hyperparathyroidism with regard to depression/anxiety, we choose to introduce the data as a narrative review since various levels of statistical evidence are identified in the mentioned papers. On the other hand, a systematic review pinpoints a specific critical assessment which in this matter with respect to the primary hyperparathyroidism is rather limited so far. However, this type of review is a well-recognized, standard, traditional approach which is suitable for topics with less generous publications such as the update of the most recent data on primary hyperparathyroidism and depression/anxiety. This allowed us to examine and evaluate the scientific panel on this specific topic in a useful way for various practitioners. Thank you
The authors did a good job of extracting relevant quantitative information. However, the authors really missed an opportunity to aggregate that data to present into summary statistics - e.g. a form of metadata analysis (which is different from meta-analysis, which would include a meta-regression equation). For example, the co-presence of depression or anxiety could have been calculated using aggregated data and presenting an odds ratio or risk ratio. Another option that is often done to aggregate diverse data in reviews such as this is to measure the association(s) of interest by calculating a Cohen's D.
Thank you very much. As mentioned, this is narrative review; hence, the statistics is used according to the original data. We introduced at Adnexa a synopsis with the most important results. Thank you.
The authors need to improve the display items. Right now they have very long and difficult to navigate tables. Inclusion of longer tables could be done using the Appendix, which would still allow it to be included with the main paper file download the way MDPI does their formatting. But, more importantly, the authors need to devise a better way to summarize the information using either a more compact table, or preferably, with the use of a figure (pie chart, bart chart, etc.).
Thank you very much. According to your recommendation, we revised the data, reduced the length of the article and moved all the tables as Adnexa (except Table 1).
Moreover, we added a synopsis with the most important results.
Notably, to our best knowledge, this is the most complex analysis that we could identify in the field of primary hyperparathyroidism and its association with depression or anxiety. Also, this is the result of a massive team work in order to identify the complex issues, pitfalls, and challenges in the mentioned area. Moreover, this is a double perspective: the one of the specialist that usually treats patients with endocrine (parathyroid) tumors (endocrinology and endocrine surgery), but, also, from the point of view of the practitioners that might involve patients confirmed with depression or anxiety, including primary health care. Hence, the complexity of the work involves a certain length of the paper. Of note, this is no length restriction according to MDPI rules, but a mandatory minimum number of words for reviews.
Thank you.
Thank you very much.

Reviewer 3 Report
Comments and Suggestions for Authors
The manuscript entitled An analysis of primary hyperparathyroidism in association with depression or anxiety is a narrative review. The authors analyzed the most recent findings regarding the link between depression and/or anxiety in subjects confirmed with primary hyperparathyroidism, including the impact of the parathyroidectomy in improving the outcome of these conditions. They concluded that the optimal method of depression/anxiety screening in primary hyperparathyroidism remains to be determined and the management should be refined upon depression/anxiety identification.
The subject is interesting and in the same time is difficult to be screened in this specific population. Therefore, it deserves to be studied.
The manuscript is well written; data were well synthetized. The analysis of literature data was well presented and well discussed.
Author Response
Response to Review 3 Comments
Dear Reviewer,
Thank you very much for your time and your effort to review our manuscript.
We are very grateful for your insightful comments and observations, also, for providing your valuable feedback on the article.
Here is a point-by-point response and related amendments that have been made in the manuscript according to your review (marked in yellow color).
The manuscript entitled An analysis of primary hyperparathyroidism in association with depression or anxiety is a narrative review. The authors analyzed the most recent findings regarding the link between depression and/or anxiety in subjects confirmed with primary hyperparathyroidism, including the impact of the parathyroidectomy in improving the outcome of these conditions. They concluded that the optimal method of depression/anxiety screening in primary hyperparathyroidism remains to be determined and the management should be refined upon depression/anxiety identification.
Thank you very much.
The subject is interesting and in the same time is difficult to be screened in this specific population. Therefore, it deserves to be studied.
Thank you very much.
The manuscript is well written; data were well synthetized. The analysis of literature data was well presented and well discussed.
Thank you very much.

Round 2
Reviewer 2 Report
Comments and Suggestions for Authors
The authors have made formatting revisions to improve the paper. Most notably, a flowchart of the review design was included and tables were re-formatted for reader clarity, better summarization, and to better meet MDPI formatting guidelines.
It is [now] understood that that this is not a "systematic" review in the sense of the formal definition. Nonetheless, non-systematic reviews should still have straightforward and transparent methodology. The authors need to improve some of the the methodology to increase credibility. The authors need to state how the review was completed to get from 929 articles to 16 articles. Were PubMed filters (existing or custom) used to remove the specific groups excluded or were exclusion done manually by human review? If done manually, were exclusions done by reviewing abstracts or full-text publications? If done manually by human review, please list the domain expertise and number of persons doing the review. Notably, some of this info was listed in the author responses, presumptively as part of claims to bolster impact and innovation. Some of the claims could be reviewed and added to the paper - particularly in the Methods. Claims on the novelty of the interdisciplinary review framework (even if most is based on the stated combination of reviewer expertise), itself, could be optionally added at the end of the Introduction in the last paragraph.
I would recommend bringing Figure 1 (flowchart of review design) back into the main paper, which is pretty standard for this journal and, more importantly, is better for readers.
While the authors responded with their desire not to perform any statistical analysis, they have already done the harder work, which was extracting the data required and putting it in tabular format. Doing even a single calculation to quantify aggregate association of depression/anxiety with primary hyperparathyroidism on the samples collected in the tables would take only a few minutes, but it would greatly increase the impact of the large data extraction summaries the authors' already performed. Obviously adding an aggregate statistic is not required for a non-systematic review, but it would greatly enhance perceived value and likely increase citations for the article. The suggested statistic in mind would be calculating a simple ratio to assess the presence of the association.
MINOR:
Given the article is a review and not introducing novel methodology or mathematical proofs that warrant first person language throughout, it would be recommended to standardize tone to third person. If necessary, a "we" in the Methods would be acceptable.
Author Response
Response to Review 2 Comments – second round
Dear Reviewer,
Thank you very much for your time and your effort to review our manuscript for the second round.
We are very grateful for providing your valuable feedback on the article.
Here is our response and related amendment that has been made in the manuscript according to your review (marked in yellow color).
The authors have made formatting revisions to improve the paper. Most notably, a flowchart of the review design was included and tables were re-formatted for reader clarity, better summarization, and to better meet MDPI formatting guidelines.
Thank you very much.
It is [now] understood that that this is not a "systematic" review in the sense of the formal definition. Nonetheless, non-systematic reviews should still have straightforward and transparent methodology. The authors need to improve some of the methodology to increase credibility. The authors need to state how the review was completed to get from 929 articles to 16 articles. Were PubMed filters (existing or custom) used to remove the specific groups excluded or were exclusion done manually by human review? If done manually, were exclusions done by reviewing abstracts or full-text publications? If done manually by human review, please list the domain expertise and number of persons doing the review. Notably, some of this info was listed in the author responses, presumptively as part of claims to bolster impact and innovation. Some of the claims could be reviewed and added to the paper - particularly in the Methods. Claims on the novelty of the interdisciplinary review framework (even if most is based on the stated combination of reviewer expertise), itself, could be optionally added at the end of the Introduction in the last paragraph.
Thank you very much. According to your recommendations, we provided more data at Methods, we expanded the statement at the end of Introduction section, while the authors’ contribution has already been mentioned in the distinct section amid MPDI format. Thank you
I would recommend bringing Figure 1 (flowchart of review design) back into the main paper, which is pretty standard for this journal and, more importantly, is better for readers.
Thank you very much. According to your recommendations, we moved it into the main paper. Thank you
While the authors responded with their desire not to perform any statistical analysis, they have already done the harder work, which was extracting the data required and putting it in tabular format. Doing even a single calculation to quantify aggregate association of depression/anxiety with primary hyperparathyroidism on the samples collected in the tables would take only a few minutes, but it would greatly increase the impact of the large data extraction summaries the authors' already performed. Obviously adding an aggregate statistic is not required for a non-systematic review, but it would greatly enhance perceived value and likely increase citations for the article. The suggested statistic in mind would be calculating a simple ratio to assess the presence of the association.
Thank you very much. At Adnexa, we provided the main findings in terms of core data (numbers). Since this is not a systematic review, any original statistics, while not being mandatory, is a matter of bias since, as mentioned, different methods/scales have been used to define depression/anxiety and a heterogeneous spectrum of endocrine population with different levels of severity with respect to the parathyroid conditions, hence, novel bias might be generated and this is out of the scope of the present work. Thank you
MINOR: Given the article is a review and not introducing novel methodology or mathematical proofs that warrant first person language throughout, it would be recommended to standardize tone to third person. If necessary, a "we" in the Methods would be acceptable.
Thank you very much. According to your recommendation, we only used third person, except for the data that refer to our findings by applying the mentioned Methods. Thank you
Thank you very much.
